# Comparing Parameters of Motor Potentials Recordings Evoked Transcranially with Neuroimaging Results in Patients with Incomplete Spinal Cord Injury: Assessment and Diagnostic Capabilities

**DOI:** 10.3390/biomedicines11102602

**Published:** 2023-09-22

**Authors:** Katarzyna Leszczyńska, Juliusz Huber

**Affiliations:** 1Department of Pathophysiology of Locomotor Organs, Poznan University of Medical Sciences, 28 Czerwca 1956 No 135/147, 60-545 Poznań, Poland; juliusz.huber@ump.edu.pl; 2Department of Neurosurgery, Wroclaw Medical University, Borowska 213, 50-556 Wroclaw, Poland

**Keywords:** incomplete spinal cord injury, electromyography, motor-evoked potentials, spinal cord neuroimaging, correlations

## Abstract

This study aimed to investigate the relationships between the different levels and degrees of incomplete spinal cord injury (iSCI) evaluated with magnetic resonance imaging (MRI) and the results of non-invasive electromyography (mcsEMG), motor-evoked potentials (MEP), and electroneurography (ENG). With a focus on patients with injuries at four different levels, C3–C5, C6–Th1, Th3–Th6, and Th7–L1, this research delved into the intricate interplay of spinal circuits and functional recovery. The study uses MEP, EMG, and ENG assessments to unveil the correlations between the MEP amplitudes and the MRI injury scores. We analysed data from 85 iSCI patients (American Spinal Injury Association—ASIA scale; ASIA C = 24, and D = 61). We compared the MRI and diagnostic neurophysiological test results performed within 1–2 months after the injury. A control group of 80 healthy volunteers was examined to establish reference values for the clinical and neurophysiological recordings. To assess the structural integrity of spinal white and grey matter on the transverse plane reconstructed from the sagittal readings, a scoring system ranging from 0 to 4 was established. The spinal cord was divided into two halves (left and right) according to the midline, and each half was further divided into two quadrants. Each quadrant was assessed separately. MEP and EMG were used to assess conduction in the corticospinal tract and the contraction properties of motor units in key muscles: abductor pollicis brevis (APB), rectus abdominis (RA), rectus femoris (RF), and extensor digitorum brevis muscles (EXT). We also used electroneurography (ENG) to assess peripheral nerve conduction and to find out whether the changes in this system significantly affect patients’ scores and their neurophysiological status. The study revealed consistent positive correlations in iSCI patients between the bilateral decrease of the spinal half injury MRI scores and a decrease of the transcranially-evoked MEP amplitudes, highlighting the complex relationship between neural pathways and functional outcomes. Positive correlations are notably pronounced in the C3–C5, C6–Th1, and Th3–Th6 subgroups (mostly r_s_ 0.5 and above with *p* < 0.05), while Th7–L1 presents distinct patterns (r_s_ less than 0.5 and p being statistically insignificant) potentially influenced by unique structural compensation mechanisms. We also revealed statistically significant relationships between the decrease of the cumulative mcsEMG and MEP amplitudes and the cumulative ENG scores. These insights shed light on the multifaceted interactions between spinal cord injury levels, structural damage, neurophysiological measures, and motor function outcomes. Further research is warranted to unravel the intricate mechanisms driving these correlations and their implications for enhancing functional recovery and the rehabilitation algorithms in patients with iSCI.

## 1. Introduction

A spinal cord injury (SCI) is a critical health condition that frequently leads to significant morbidity and long-lasting impairment. It represents a complex challenge with profound implications for motor function. There are different levels of damage and different mechanisms of injury that lead to different severity of symptoms. A common feature of such injuries is the axons within the white matter spinal cord funiculi that are interrupted, leading to disturbances of sensory and motor capabilities below the level of injury. Typically caused by major trauma, mostly vehicle accidents and acts of violence, the initial damage is often irreversible [1]. On a global scale, an estimated 250,000 to 500,000 individuals experience spinal cord injuries annually [1,2]. According to the Global Burden of Disease Study 2019 done by Ding et al. (2022), the occurrence and impact of SCI have risen in the past three decades. This is a tremendous group of primarily young people who, if they survive an injury, have to face the consequences of it for the rest of their lives. Therefore, developing not only treatment methods but also algorithms for diagnosing and evaluating patients is key to handling the increasing number of individuals affected by spinal cord injury [2]. Furthermore, both neurosurgeons and neurologists, relying upon the clinical investigations incorporated within the assessment of the American Spinal Injury Association (ASIA) Impairment Scale, occasionally express apprehension regarding the prognostication of therapeutic efficacy. This concern pertains to the selection of therapeutic modalities, or the ongoing pursuit of interventions initiated by patients and their familial caregivers.

The acute and subacute phases of the SCI are the immediate periods following the injury. They are typically defined as the first 24 to 48 h and days to the first few months after the trauma. Prompt and appropriate management is crucial during these critical phases to minimise further damage and optimise the outcomes. One key step is obtaining diagnostic imaging, such as magnetic resonance imaging (MRI). Evaluating the extent and location of the spinal cord injury and guiding appropriate treatment decisions is essential. Initiating measures to prevent or minimise secondary injury is also critical. This includes maintaining spinal alignment, avoiding hypotension or hypoxia, and providing immediate stabilisation and immobilisation of the spine to prevent additional spinal cord damage [3,4]. In some cases, surgical interventions may be necessary for the acute phase of iSCI. These interventions aim to stabilise the spine, decompress the spinal cord, and manage associated injuries, such as fractures or dislocations. The decision for surgical intervention should be made on a case-by-case basis, considering factors such as the level and severity of the injury, instability, neurological deficits, and associated injuries [3,5,6]. The time for awaiting the results of traditional rehabilitation by means of the targeted patient’s personal kinesio- and physicotherapy is usually determined by the oedema associated with most of cases of iSCI which last more than a year, limiting the advances of treatment results [7,8,9].

Surgical stabilisation after a spinal cord injury should ideally be performed as soon as possible to maximise the chances of neurological recovery and prevent further damage to the spinal cord [3,4,5,6]. The frequent use of metallic stabilisation hardware in SCI management can lead to artefacts in post-surgical MRI assessments rendering them difficult to interpret [6]. Hence, the implantation of surgical stabilisation elements immediately following spinal injury poses challenges in assessing the structural MRI of the affected spinal cord regions. Reliable evaluations are attainable solely through MRI scans acquired with devices possessing a high resolution of 3–5 Tesla, limited to the preoperative assessment immediately after the injury event. The subsequent evaluations are hindered by image distortions and artefacts arising from the interaction between titanium stabilisation and the magnetic resonance field. Moreover, neurosurgeons express a keen interest not only in the structural condition but also in the functional state of white matter cord conductivity. A functional assessment of spinal cord conduction capacity is also of great importance for rehabilitation algorithms and the evaluation of patient progress during further therapy. This can be assessed directly through motor-evoked potential testing (MEP) and indirectly via the evaluation of electromyography recordings (EMG) from muscles innervated by neuromeres at the location of spinal cord injury and below.

Both MRI and MEPMEP are valuable tools for understanding the extent and nature of spinal cord injuries, but they provide different types of information. MRI offers detailed anatomical images, while MEPMEP provide functional information about the integrity of motor pathways. Rare attempts have been made to establish a correlation between the assessment of MRI and MEP in patients with iSCI, however, this correlation has not been previously explored. Thus, we sought to address this methodological gap by identifying a straightforward and dependable approach. The null hypothesis posits that no direct correlation can be identified between the MRI and MEP results in iSCI patients with injuries spanning four distinct levels: C3–C5, C6–Th1, Th3–Th6, Th7–L1. However, if such a correlation does exist, and if motor function outcomes in iSCI patients, particularly those with lower thoracolumbar injuries, surpass those in patients with cervicothoracic injuries, it could suggest that the neuronal circuits within the spinal cord compensate for the impaired efferent transmission.

The spinal cord possesses intricate neural circuits and plasticity mechanisms that might contribute to functional compensation after injury. Numerous neurophysiological experimental and clinical investigations have indicated the involvement of the propriospinal system in the spinal cord in the process of compensating for neuroplasticity following iSCI [10,11,12]. The outcomes of this study could potentially provide a basis for considering this as a plausible mechanism.

In this study, we decided to compare the results of a clinical neurophysiology test—non-invasive electromyography recorded with the surface electrodes (sEMG) and MEP in iSCI patients and healthy people—to show the extent of the pathology. But more importantly, we juxtaposed data from clinical neurophysiology studies assessing spinal cord function with the MRI findings assessing the spinal cord structure in patients with iSCI damage at four different levels of neuromeres, shedding light on the complex relationships within spinal cord neural networks. Our goal has been to answer if neurophysiological tests can serve as a supplementary method to an MRI structural assessment in order to enhance the evaluation of the severity of spinal cord injury, and if there is a correlation between the assessment of the MRI and MEP in patients with iSCI. Understanding these relationships is vital for a functional evaluation of the spinal cord function by the neurosurgeon or neurologist, tailoring effective rehabilitation strategies, and optimising patient care.

## 2. Materials and Methods

### 2.1. Patients and Healthy Subjects

In this study, we have analysed the data from 85 iSCI patients (American Spinal Injury Association—ASIA scale; ASIA C = 24, and D = 61), 1.3 ± 0.3 months after C3–L1 spinal injury (Table 1). Unlike our previous research projects [7,8,13], in this study, we compared the MRI images and diagnostic neurophysiological tests results performed within 1–2 months after the injury.

The patient selection process considered various criteria, including the level of spinal cord injury, similarity in the degree of injury based on neuroimaging results (preservation of 1/3 to 1/4 of the spinal cord structure), age, gender, and the average time elapsed since the trauma. Four groups were formed based on the injury levels: C3–C5, C6–Th1, Th3–Th6, and Th7–L1. Neurophysiological tests were conducted, encompassing surface electromyography during maximal contraction (mcsEMG) in the specific key muscles for the trunk (rectus abdominis), upper extremity (abductor pollicis brevis), and lower extremities (rectus femoris and extensor digitorum brevis). Electroneurography (ENG) was also performed to evaluate peripheral motor impulse transmission in the median and peroneal nerves. All patients included in the study were in the early stages of their post-injury assessment. Those who had sustained the injury more than two months prior to enrollment were excluded. Additional exclusion criteria included electronic implants/devices (e.g., pacemakers, insulin pumps, baclofen pumps, cochlear implants), stroke, plexopathies, mononeuropathies and polyneuropathies, inflammatory diseases, confirmed COVID-19-related disorders, and myelopathies diagnosed before or after the incident. Informed consent was obtained from all participants, and the study adhered to the principles of the Declaration of Helsinki, with approval from the Bioethics Committee of the Medical University (decision no. 942/2021).

A control group of 80 healthy volunteers was examined to establish reference values for clinical and neurophysiological recordings. To ensure comparability, the control group’s demographics (gender, age, height, and weight) were adjusted to match those of the study group. Statistically significant differences in age, height, and weight between the study groups and healthy volunteers in the control group were not observed (see Table 1).

All participants were aware that their involvement in the study would not yield any financial benefits, and they provided written consent for voluntary participation.

### 2.2. Key Muscles

The division of four groups of subjects with the levels of injury C3–C5, C6–Th1, Th3–Th6, and Th7–L1 was based on the innervation level of selected key muscles and the location of motor centres relative to the neuromeres innervating these muscles. The abductor pollicis brevis muscle is primarily innervated by the median nerve, which receives contributions from the spinal cord levels C8 and T1. The rectus abdominis muscle is innervated by the thoracic spinal nerves, specifically the anterior rami of the lower thoracic nerves T7 to T12. The rectus femoris muscle is part of the quadriceps muscle group, and its innervation primarily comes from the femoral nerve which receives contributions from the spinal cord levels L2 to L4. The extensor digitorum brevis muscle is primarily innervated by the deep fibular nerve (also known as the deep peroneal nerve) which receives contributions from the spinal cord levels L4 to S1. The primary innervation for the extensor digitorum brevis muscle comes from the L5 spinal cord level. These are the muscles that are standardly studied and are referred to as key muscles in clinical neurophysiology. This division into groups according to the level of damage allowed us to compare results for these muscles and to ensure that the distribution by patient size was as equal as possible.

### 2.3. Clinical Assessment Tools—MRI

MRI is a non-invasive imaging technique that can provide detailed images of the spinal cord and surrounding structures. It is often used to visualise the extent of the spinal cord damage, the location of the injury, and the presence of any abnormalities or changes in the spinal cord tissue. MRI can be useful for assessing the severity of spinal cord injuries and guiding treatment decisions [6].

In the research conducted by Ko et al., measurements of the transverse and sagittal diameters of the neuromeres from C5 to T1 ranged between 13–10 mm and 7–8 mm, respectively [14]. For the neuromeres from T6 to L1, the corresponding ranges were 7–6 mm and 6–7 mm, respectively. These dimensions provide sufficient clarity to distinguish between white and grey spinal structures. Sagittal MRI sections offer a clearer picture compared to the transverse sections, allowing for distinct identification of the border between white and grey matter. By examining sagittal sections from left to right with approximately 2 mm dimensions, as typically performed in diagnostic imaging tests, it becomes possible to estimate the homogeneity of the dorsal, lateral, and ventral funiculi along their greatest dimension and to observe the central canal of the spinal cord.

For the analysis, only data from patients with cervical or thoracic level damage were selected. These patients had MRI images of at least 3T, with five sagittal sections assessable from left to right. To assess the structural integrity of the spinal white and grey matter on the transverse plane reconstructed from the sagittal readings, a scoring system ranging from 0 to 4 was established. The spinal cord was divided into two halves (left and right) according to the midline, and each half was further divided into two quadrants (Figure 1C). Evaluating each quadrant separately during a sagittal inspection from left to right through the midline, where the line of the central canal is visible (Figure 2), allowed for the creation of a spinal half-injury MRI score. A score of 1 represented normal spinal cord integrity. A score of 0.5 indicated partial injury, signifying some structural compromise. A score of 0 indicated total injury, suggesting complete loss of structural integrity. Each evaluation of the spinal halves divided into two quadrants contributed 4 points, while one spinal half received 2 points, and each quadrant was given 1 point. This type of system is innovative and lacks direct references to the literature.

### 2.4. Neurophysiological Testing Methods—MEP, EMG, ENG

In the study, we used two main methods of neurophysiological assessment (MEP, EMG) and a third (ENG) to support the study’s methodology. MEP and EMG were used to assess conduction in the corticospinal tract and contraction capacity in the muscle tissue. We used ENG to assess peripheral nerve conduction to evaluate whether the changes in this system significantly affect patients’ scores and their neurophysiological status. Figure 1 illustrates the methodological tenets of the neurophysiological investigations. The assessments were conducted within a climate-controlled environment maintained at an average temperature of 22 °C.

MEP are neurophysiological tests used to assess the integrity of the motor pathways in the spinal cord. MEP involve delivering transcranial magnetic stimulation to the motor cortex of the brain, which then travels down the motor pathways to activate muscles in the body. By recording the resulting muscle responses, healthcare providers can evaluate the conduction of signals along the motor pathways and identify potential disruptions or abnormalities. MEP were elicited through transcranial magnetic single stimulus (TMS) utilising a magnetic circular coil (C-100, 12 cm in diameter). The coil was positioned over the scalp within the M1 motor cortex region, targeting the corona radiata excitation angle. This area corresponds to the origin of the corticospinal tract for the upper and lower extremities. Surface electrodes were placed on the key muscles bilaterally for recording. The MagPro X100 magnetic stimulator (Medtronic A/S, Skovlunde, Denmark) was employed for MEP testing. The magnetic field produced by the coil, with a strength of 70–80% of the resting motor threshold (RMT; 0.84–0.96 T), penetrated neural structures up to 3–5 cm in depth. Primary outcome measures encompassed the analysis of latency and the amplitude parameters to evaluate the output of the primary motor cortex and assess the transmission of neural impulses to the effectors via spinal cord descending tracts. A sequence of successive tracking attempts determined the optimal stimulation location (hot spot) where the largest MEP amplitude was elicited by TMS, with a 5 mm distance between each attempt. Amplitude was gauged from peak to peak of the signal, while latency was measured from the stimulus application, marked by the artefact in the recording, to the onset of the positive inflexion of potential. Participants, both patients and healthy volunteers, experienced the stimulation as non-painful, although a mild sensation of current spreading to the lower extremities was noted. Participants remained awake and cooperative throughout the procedure. MEP were recorded using the 8-channel KeyPoint Diagnostic System (Medtronic A/S, Skovlunde, Denmark), and standard disposable Ag/AgCl surface electrodes with a 5 mm^2^ active surface were employed. The recorder’s settings encompassed a low-pass filter at 20 Hz, a high-pass filter at 10 kHz, a time base of 10 ms/D, and a signal amplification set between 200–5000 µV. Recordings were digitised at a rate of 2000 samples per second and channel, within a bandwidth of 10 Hz to 1000 Hz. The electroconductive gel was applied to minimise resistance between the electrode surface and the skin.

Bilateral surface electromyography (sEMG) recordings were executed focusing on key muscles. These recordings aimed to evaluate the motor unit recruitment during a 5-s maximal contraction endeavour (Figure 1B,C). Employing the KeyPoint Diagnostic System (Medtronic A/S, Skøvlunde, Denmark), sEMG recordings were conducted on supine patients within the examination context. For sEMG measurements, we employed standard, disposable Ag/AgCl surface recording electrodes with an active surface area of 5 mm^2^. These were positioned with precision: an active electrode placed over the muscle belly, a reference electrode positioned on the distal tendon of the same muscle, and a grounding electrode placed on the distal region of the examined muscle, adhering to the protocols outlined by the International Federation of Clinical Neurophysiology—European Chapter, the ones used in our department [8,13,15,16]. The recorder settings included upper and lower filter limits set at 10 kHz and 20 Hz, respectively. Participants were instructed to engage in maximal muscle contractions for 5 s upon the examiner’s command. Each session encompassed three attempts interspersed with 1-min intervals for rest. The analysis focused on the optimal attempt, defined by the highest mean amplitude measured peak-to-peak in relation to the isoelectric line. Outcome parameters consisted of amplitude measurements in μV and the frequency of motor unit action potential recruitment in Hz. Based on the calculations of motor unit action potential recruitment during maximal contraction, a frequency index (FI, ranging from 3 to 0) was assigned: 3 denoting 95–70 Hz (normal), 2 for 65–40 Hz (moderate abnormality), 1 representing 35–10 Hz (severe abnormality), and 0 indicating no contraction. sEMG recordings for both control subjects and patients were executed at a time base of 80 ms/D and amplification levels between 20 and 1000 μV/D.

Bilateral electroneurography (ENG) was employed to scrutinise neural impulse transmission in the motor peripheral fibres of the peroneal nerves. The objective was to discern potential variations in nerve conduction that could impact muscle function evaluation and the efferent transmission measurements. The procedure entailed delivering rectangular pulses of 0.2 ms duration at a frequency of 1 Hz and an intensity ranging from 0 to 80 mA, using bipolar stimulating electrodes situated over the skin along the anatomical paths of the nerves at the ankle. Compound muscle action potentials (CMAP) and F-waves were then captured from the extensor digitorum brevis muscles. These recordings verified neuronal impulse transmission in motor peripheral fibres and within L5 ventral spinal roots, respectively. Settings for these recordings included amplification levels between 500 and 5000 µV/D and a time base of 5–10 ms/D. The results were then compared against normative values established in the healthy volunteers. The parameters of interest encompassed amplitudes (in µV) and latencies (in ms) for M-waves, inter-latencies of recorded M-F waves (in ms), and frequencies for F-waves (normally not below 14 during the evocation of 20 successive recordings of M-waves). The measurements were performed at an amplification of 5–5000 µV and a time base of 2–10 ms. A comparison was drawn between the obtained results from the patients and the established normative values in the healthy volunteers, building upon methodologies described in previous works authored by members of our research team [13,16].

### 2.5. Statistical Analysis

We analysed the data using Statistica, version 13.1 (StatSoft, Kraków, Poland). Descriptive statistics, such as mean values, standard deviations (SD), and minimum and maximum values, were calculated for measurable variables. Data mining was conducted to match the patients and healthy volunteers based on age, sex, and basic anthropomorphic properties like weight and height. We assessed the normality distribution and homogeneity of variances using Shapiro-Wilk tests and Levene’s tests. For both the healthy subjects (controls) and the patients with incomplete spinal cord injuries at the cervical and thoracic levels, we compared the mean values of the parameters obtained from the sEMG, ENG, MEP, and the clinical tests using a Student’s t-test and the Mann-Whitney test. We considered a significance level of *p* < 0.05 to indicate significant statistical differences. To determine the required sample size, we performed a preliminary statistical analysis using primary outcome variables from the sEMG and MEP recordings of the extensor digitorum brevis muscles, with a power of 80% and a significance level of 0.05 (two-tailed). We calculated the mean and standard deviation (SD) using data from twenty subjects in both the patient and the healthy control groups. According to the sample size software, we estimated that a minimum of 60 subjects from each group were needed. We used the non-parametric Spearman’s rank correlation coefficient (r_s_) to examine correlations between MEP measurements and the MRI results. We considered a significance level of *p* < 0.05 to indicate a statistically significant rank correlation.

## 3. Results

Table 2 provides a comprehensive overview of the electroneurography outcomes within the cohort of healthy volunteers (referred to as controls) as well as in each respective patient group. Within the control group, all participants exhibited unremarkable motor peripheral transmission of neural impulses along the median and peroneal nerve fibres. Notably, the patient cohorts revealed a prevailing presence of peripheral neural transmission abnormalities, primarily manifesting as levels closely approximating severe and moderate deviations from the norm. The assigned scores ranged from 1.3 to 1.7 among patients with iSCI spanning from C6 to L1, while in the C3 to C5 group, the score reached 2.1. It is important to note that a score of 3 signifies normal transmission. A comparative analysis revealed statistically significant deviations in the ENG scores between patients and controls, with *p*-values ranging from 0.04 to 0.03.

Table 3 presents a detailed and intricate analysis that compares the measurements obtained from the MEP and mcEMG assessments on both the left and right sides. This comprehensive examination encompasses distinct patient subgroups and includes comparisons with healthy participants. Notably, this table also introduces a novel aspect by integrating spinal half-injury MRI scores, a novel metric designed to evaluate the integrity of white matter axons spanning the spinal cord. The approach involves segmenting the cord cross-section into four distinct parts.

Across the diverse muscles under scrutiny, the activity of motor units and the conductivity characteristics of the efferent pathways exhibited distinct patterns. Specifically, the patients classified within the C3–C5 subgroup (illustrated in Figure 2A, right portion) exhibited substantial bilateral impairments when juxtaposed with the control cohort. These disparities held significant statistical value, marked by *p*-values within the range of 0.04 to 0.009. Parallel findings emerged in the C6–Th1 iSCI subgroup, manifesting predominantly within the lower extremity muscle recordings (depicted in Figure 2B, right portion). Intriguingly, the Th3–Th6 subgroup displayed remarkable abnormalities, with a heightened occurrence in lower muscles as opposed to the rectus abdominis (RA) muscles (shown in Figure 2C, right portion). Conversely, the patients within the Th7–L1 subgroup demonstrated positive outcomes in terms of the mcsEMG parameters (more amplitudes than FI indexes) and the MEP parameters, particularly in recordings from the upper extremities (depicted in Figure 2C, right portion). The parameters extracted from sEMG and MEP recordings for Th3-Th6 iSCI patients exhibited similarities to those of the healthy counterparts, their MRI injury scores were notably more adverse.

In the lower-right quadrant of Table 3, the data consistently reveal a conspicuous trend. Specifically, the ventral regions of the spinal cord consistently display higher MRI injury scores, indicative of a more significant structural compromise. This observation is particularly salient within the C3–C5 and C6–Th1 iSCI subgroups when juxtaposed with the Th3–Th6 and Th7–L1 groups.

Among all patients with incomplete spinal cord injuries (iSCI), we identified consistent and robust positive correlations. These correlations were observed between the reduction in amplitudes obtained from both upper and lower extremity muscles during surface electromyography (sEMG) assessments and the corresponding amplitudes of motor- evoked potentials (MEP). Likewise, there were analogous associations with the amplitudes of M-waves elicited by the peroneal nerve stimulations during electroneurography (ENG) measurements and the MEP recordings from the muscles associated with extension (EXT) actions. These findings are comprehensively presented in Table 4.

The most noteworthy instances of positive correlations, as measured by the r_s_ coefficients, were discerned between the MRI injury scores and the MEP amplitudes recorded on both the left and right sides. These significant correlations were particularly pronounced within the subgroups of iSCI patients categorised as C3–C5 and C6–Th1. A similar pattern emerged in Th3–Th6 iSCI patients, where the robust r_s_ coefficients depicted correlated relationships on both sides of the body. In these instances, positive correlations were evident between the MEP amplitudes and the amplitude parameters for both rectus abdominis (RA) muscles and lower extremity muscles. Notably, within the Th7–L1 iSCI group, the positive correlations were specifically confined to EXT muscles.

However, in comparing the iSCI Th7–L1 subgroup with the Th3–Th6 patients, a significant distinction becomes apparent. Specifically, the former group does not manifest uniformly high or very high positive correlation coefficients between the MEP amplitude parameters and the low scores indicative of spinal half-injury on the MRI scans.

## 4. Discussion

The study reveals consistent positive correlations in iSCI patients between the bilateral decrease of the spinal half injury MRI scores and a decrease of transcranially-evoked MEP amplitudes, shedding light on the intricate relationships between muscle function, neural signalling, and structural integrity. These findings underscore the complex interplay between neural pathways, muscle responses, and the anatomical consequences of spinal injuries.

The general principle is that the higher the level of spinal cord injury, the more extensive the impairment is likely to be [17]. It is important to remember that these are general observations, and that the actual EMG and MEP recordings can vary greatly among individuals within each group. Additionally, the extent and severity of the injury, as well as other individual factors, such as clinical conditions before the injury, can influence the outcomes [18]. The two groups, Th3–Th6 and Th7–L1, typically result in muscle weakness affecting the lower extremities. The upper extremities are generally unaffected. The EMG recordings may show preserved muscle activity in the upper limbs, similar to the control group, which was confirmed in our study by the lack of statistical significance in the MEP results in the APB muscle between the healthy controls and the patients in the two groups (see Table 4) [18,19,20].

Table 3 offers a comprehensive exploration of the comparative aspects of MEP and mcEMG testing outcomes across diverse patient subgroups and healthy individuals. Importantly, the integration of spinal half-injury MRI scores enriches our understanding of white matter axon continuity within the cross-sectional configuration of the spinal cord. These findings underscore the intricate interplay of neural efferent transmission in the intricate context of spinal cord injuries, evidenced by the distinctive abnormalities noted in the various patient clusters. Notably, while the parameters extracted from the sEMG and MEP recordings for Th3–Th6 iSCI patients exhibited similarities to those of the healthy counterparts, their MRI injury scores were notably more adverse. This discrepancy hints at the potential presence of structural elements that could potentially compensate for abnormal neural efferent transmission, thereby influencing the control of spinal motor centres below the injury level.

Propriospinal neurons are a specific type of nerve cells that play a crucial role in the communication between different segments of the spinal cord. They are interneurons that connect various levels of the spinal cord, allowing for coordinated movement, sensory integration, and neural transmission. These neurons receive sensory information from peripheral nerves and transmit signals to the motor neurons, influencing voluntary and reflexive movements. Propriospinal neurons are primarily located within the grey matter of the spinal cord, spanning multiple segments. They extend their axons both rostrally (towards the head) and caudally (towards the feet) over considerable distances, forming long-range connections within the spinal cord. Propriospinal neurons exhibit a certain degree of plasticity, meaning they can undergo structural and functional changes in response to injury [21,22]. This plasticity can include the sprouting of new connections, a reorganisation of synaptic connections, and changes in the expression of neurotransmitters and receptors. Consequently, propriospinal neurons may compensate for the loss of descending input from the brain by forming new connections with the neighbouring intact segments. These adaptive changes can potentially contribute to some degree of functional recovery or compensation after SCI. These changes may contribute to the preservation or recovery of some motor functions. In the mcsEMG recordings, the activity of muscles innervated by the propriospinal pathways may be detectable, potentially indicating preserved or recovered motor control despite the injury [21,22,23,24,25]. Similarly, the MEP recordings may show motor responses in muscles influenced by the propriospinal pathways, suggesting the presence of alternative neural pathways supporting motor function. Propriospinal neurons facilitate intersegmental communication within the spinal cord. This coordination can influence motor output patterns. In EMG recordings, the presence of appropriate intersegmental coordination, albeit possibly impaired, may suggest some preserved communication between different levels of the spinal cord via propriospinal pathways. The MEP recordings may also show motor responses that involve multiple spinal cord segments, indicating the presence of intersegmental coordination mediated by the propriospinal connections.

Considering the distribution of spinal pathways (Figure 1C), it is reasonable to assume that the dorsolateral quadrants I and II contain more efferent white matter funiculi fibres than the afferent ones, while ventral III and IV show the opposite pattern. Consequently, damage to a specific quadrant would be expected to result in specific motor deficits (involving lateral cortico- and rubrospinal pathways) rather than sensory deficits during the MEP recordings. The anterior corticospinal tract fibres and long propriospinal descending and ascending fibres, located in close proximity to the medial part of the ventral horn and the central canal, form another efferent system of white matter fibres transmitting commands from the motor cortex. In this study, only the motor component’s function was directly assessed through MEP and indirectly through the bilateral EMG recordings.

Hypothetically, when recording MEP from the rectus abdominis muscle and the rectus femoris muscle with a unilateral lesion at the C5–T1 level, comparable MEP responses may be observed on the right and left sides. However, with a lesion at the T6–L1 level (i.e., after damage to the propriospinal neurons), the MEP recordings on the side of the lesion may be worse, indicating damage to the propriospinal system (Th7–L1) compensating for efferent conduction to the muscles of the lower limbs (Figure 1B).

A very high positive Spearman correlation indicates a strong relationship between two variables: as one variable increases, the other variable tends to increase as well. Also, a strong correlation between two variables may suggest that the changes in one variable can be used to predict or explain changes in the other variable. In practical terms, the strong Spearman correlation between the MRI score and MEP indicates that there is a meaningful association between the severity of structural changes observed in MRI (such as lesions, tissue damage, or abnormalities) and the measured motor-evoked potentials parameters. This could imply that more severe structural abnormalities observed in the MRI correspond directly to neuronal excitability or responsiveness in the motor system, leading to a weaker MEP. It is important to note that most of the presented correlations are moderately strong, but not perfect, which means that while there is a discernible relationship between the MRI score and MEP, other factors may also contribute to the variations in the MEP values. Additionally, correlation does not imply causation, so the MRI score alone cannot be claimed as the cause of the MEP changes. In a clinical context, these correlations may have implications for assessing spinal cord or brain injuries or evaluating neurological disorders. For example, if the MRI score decreases in a patient with suspected spinal cord injury, it might suggest more significant structural damage and potentially be associated with altered motor responses as measured by MEP. Similarly, researchers or clinicians studying the impact of certain treatments on brain or spinal cord conditions may use this correlation to evaluate changes in the MRI, and the MEP measures over time to understand the treatment effects better. However, we have also discovered very weak correlations between the MRI scores and MEP parameters in the Th7–L1 group of patients. These correlations are also statistically insignificant so these patients present different patterns of structural and functional relationships than other groups. It suggests that there is insufficient evidence to support a significant association between the MRI score and MEP in the given data set. The findings suggest that any relationship between the MRI score and MEP is very weak and may not be practically meaningful. Other factors not considered in the study may have a more substantial impact on the MEP responses, and the MRI score alone (meaning structural assessment) may not reliably predict the MEP values that represent functional capacity. And such a hypothesis should be taken into account when planning and implementing a rehabilitation programme.

One major limitation of this study is the sample size. However, spinal cord injury is indeed a complex and relatively rare condition. Obtaining a sizable sample of patients with SCI who not only represent different injury levels but also have access to high-quality MRI and can undergo neurophysiological testing within a short period after injury has been a significant challenge. In our region, such comprehensive examinations for SCI patients are not common, which limited our ability to expand the sample size further. Another study limitation might be the 0–4 scoring system for assessing spinal white and grey matter integrity. While innovative, it lacks direct references in the literature. This scoring system was developed internally based on the specific criteria derived from high-quality MRI assessments only. Some patients were excluded from the study because their MRI data could not be assessed properly. Our primary focus was to evaluate patients during the subacute phase of spinal cord injury (SCI), which is characterized by a critical period of medical decision-making. During this phase, patients often face the necessity of surgical stabilization and the initiation of intensive rehabilitation programs. These interventions can significantly impact the patient’s baseline condition, making it essential to assess patients within this relatively short timeframe. It is important to clarify that our study did not aim to investigate the patients’ long-term outcomes after SCI. Instead, we sought to capture a snapshot of their neurophysiological status during the subacute phase, which is critical for clinical decision-making and treatment planning. This type of approach may also be considered as study limitation.

This study could serve as a basis for further research, where one could conduct studies to investigate the correlations between injury levels, motor function outcomes, and the potential role of intrinsic spinal cord circuits in compensation. It bridges the gap between the MEP recordings, the EMG parameters, and the MRI scores, unravelling the intricate relationships within incomplete spinal cord injuries. These findings pave the way for a deeper understanding of neural adaptations, facilitating targeted rehabilitation strategies and enhancing diagnostic capabilities for optimising functional recovery. However, it is important to emphasise that each patient’s response to injury and recovery is highly individual and can be influenced by numerous factors. It is important to note that iSCI patients can be in very different clinical states depending on the level of damage, the mechanism of the injury, as well as the fitness and clinical condition before the injury. Their post-injury evaluation may vary, and a comprehensive evaluation by a medical professional is necessary to assess the specific condition of each patient. Further research into assessment algorithms, as well as various rehabilitation and therapy methods, is needed. Future studies should also consider larger, more balanced sample sizes, possibly introduce additional comparison groups, and maybe try to develop a more standardized MRI scoring system after rigorous validation.

## 5. Conclusions

Among iSCI patients, robust correlations emerge between the MEP amplitude and the MRI score established for this study, highlighting the complex relationship between neural pathways and functional outcomes. Positive correlations are notably pronounced in the C3–C5, C6–Th1, and Th3–Th6 subgroups, while Th7–L1 presents distinct patterns potentially influenced by unique structural compensation mechanisms. Moreover, electromyography and transcranially-evoked motor potentials may and, in our opinion, should serve as complementary tools to enhance spinal cord injury assessment, alongside the structural evaluation in MRI in clinical studies. Understanding these relationships is vital for functional evaluation of the spinal cord function by the neurosurgeon or neurologist, tailoring effective rehabilitation strategies, and optimising patient care.

## Figures and Tables

**Figure 1 biomedicines-11-02602-f001:**
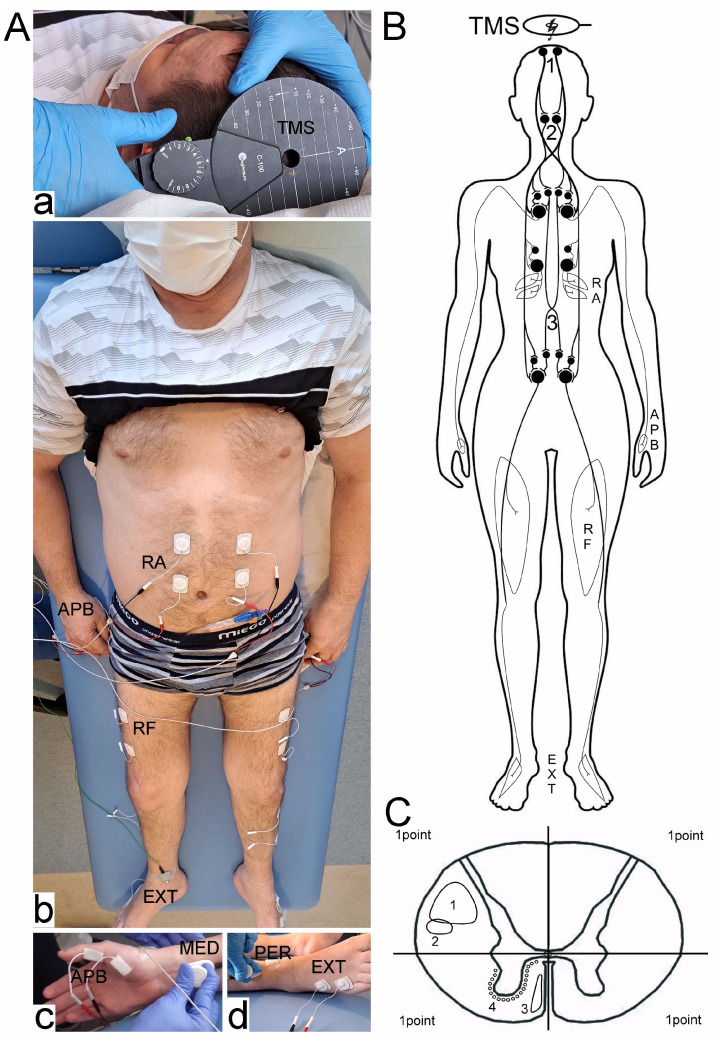
(**A**) Photographs illustrating MEP (**a**) and sEMG (**b**) recordings methodology with pairs of electrodes placed bilaterally over the surface of abductor pollicis brevis (APB), rectus abdominis (RA), rectus femoris (RF), and extensor digitorum brevis muscles (EXT) in healthy volunteers and in patients with iSCI at different spinal levels. Transcranial magnetic stimulation (TMS) was applied through the magnetic coil positioned over the skull in areas of the motor cortex innervating muscles of the upper and lower extremities. Peripheral conduction of the motor neural impulses was verified in ENG studies following the electrical stimulation at the wrist and ankle of the median (**c**, MED) and peroneal (**d**, PER) nerve branches and recordings of evoked potentials from APB and EXT muscles, respectively. (**B**) A simplified diagram of the anatomical structures transmitting the neural excitation to the motor centres of the spinal cord after transcranial magnetic stimulation of the motor cortex centres. Large black circles denote motoneurones, medium-sized—cells of origin of the descending efferent pathways, and small—interneurons. Open white areas show the location of applied bipolar surface electrodes for MEP recordings from upper, abdominal, and lower extremities muscles bilaterally. 1—corticospinal tract, 2—rubrospinal tract, 3—long descending propriospinal tract. (**C**) Transverse plane of the thoracic spinal cord presenting the idea of calculation of the spinal half injury MRI score (0–1; for each quadrant 1—normal, 0.5—partial injury, 0—total injury). Each evaluation of the spinal halves divided into two quadrants would give 4 points, for one spinal half 2 points, and for one quadrant 1 point. Location of the white matter fibres transmitting the efferent neural signals following TMS: 1—lateral corticospinal tract, 2—rubrospinal tract, 3—anterior corticospinal tract, 4—long descending propriospinal tract. Abbreviation: ⤚—excitatory synapse.

**Figure 2 biomedicines-11-02602-f002:**
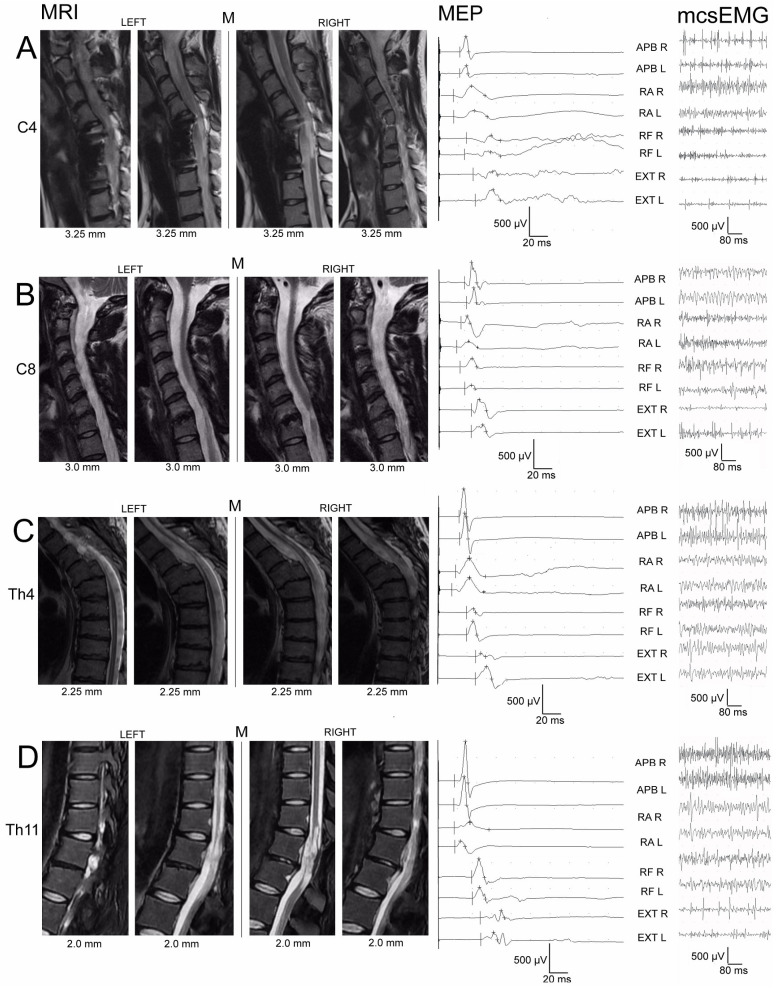
Representative examples of MEP and mcsEMG recordings in patients with iSCI at the levels of C4 (**A**), C8 (**B**), Th4 (**C**), and Th11 (**D**) neuromeres. Sequential MRI sagittal slides on the left side of the figure present the methodology of reconstruction of the spinal white and grey matter structural status allowing for the evaluation of the spinal half injury MRI score in these patients. The positions of the analysed sections in relation to the midline in the sagittal axis are expressed in millimetres. Abbreviations: APB—abductor pollicis brevis muscle; RA–rectus abdominis muscle; RF–rectus femoris muscle; EXT—extensor digitorum brevis muscle; mcsEMG—sEMG recording during maximal contraction; MEP–motor-evoked potential; R—right; L—left; M—midline.

**Table 1 biomedicines-11-02602-t001:** Demographic, anthropometric, and trauma characteristics of the iSCI patients and healthy volunteers from the control group. Minimum-maximum, mean values, and standard deviations are presented.

Subjects	Age (years)	Height (cm)	Weight (kg)	BMI	Average Time from Injury (Months)	ASIA AIS Score	Injury Level
Patients N = 85 ♀ = 19, ♂ = 66	18–64 39.3 ± 9.5	153–194 175.7 ± 7.9	49–96 61.4 ± 5.3	20.8–23.4 21.6 ± 4.2	1–2 1.3 ± 0.3	C = 24 D = 61	C3–C5 = 16 C6–Th1 = 19 Th3–Th6 = 18 Th7–L1 = 32
Control N = 80 ♀ = 17, ♂ = 63	18–60 38.1 ± 8.4	155–189 173.6 ± 6.3	45–91 60.9 ± 4.2	20.3–23.9 21.9 ± 4.1	NA	NA	NA
*p*	0.08	0.09	0.08	0.09	NA	NA	NA

Abbreviations: *p* < 0.05 determines significant statistical differences; NA—non-applicable.

**Table 2 biomedicines-11-02602-t002:** Comparison of the results from electroneurographical assessment in healthy volunteers and iSCI patients. Minimum, maximum, mean values, and standard deviations are presented.

Subjects	Injury Level	ENG Score (0–3)
Patients N = 85 ♀ = 19, ♂ = 66	C3–C5 = 16 C6–Th1 = 19 Th3–Th6 = 18 Th7–L1 = 32	1–3 2.1 ± 0.9 0–3 1.5 ± 0.9 0–3 1.7 ± 1.0 0–3 1.3 ± 0.8
Control N = 80 ♀ = 17, ♂ = 63	NA	3–3 3.0
*p*	NA	**0.04–0.03**

Abbreviations: NA—non-applicable; ENG score—electroneurographical evaluation of impulses transmission in motor fibres (0—no transmission, 1—severe abnormalities, 2—moderate abnormalities, 3—normal transmission); *p* < 0.05 determines significant statistical differences marked bold.

**Table 3 biomedicines-11-02602-t003:** Comparison of results from surface electromyography, motor-evoked potential recordings, and neuroimaging analysis results studied in a group of healthy volunteers and in the groups of patients with incomplete spinal cord injuries at different levels. Ranges, mean, and standard deviation values recorded on the left and right sides are presented.

Examined Muscle	Test	Healthy Volunteers (Control)		Patients C3–C5		Patients C6–Th1		Patients Th3–Th6		Patients Th7–L1		*p* vs. Patients C3–C5		*p* Control vs. Patients C6–Th1		*p* Control vs. Patients Th3–Th6		*p* Control vs. Patients Th7–L1	
		Right	Left	Right	Left	Right	Left	Right	Left	Right	Left	Right	Left	Right	Left	Right	Left	Right	Left
APB	mcsEMG Amplitude (µV)	900–5500 2725.1 ± 115.3	900–600 2720.0 ± 109.2	50–1000 350.2 ± 174.1	50–1000 425.7 ± 130.2	50–2500 835.6 ± 110.1	50–2500 830.4 ± 119.4	200–4000 2450 ± 261.3	300–4000 2010.1 ± 235.5	800–5000 2088.1 ± 134.8	800–4500 2232.1 ± 193.3	**0.009**	**0.007**	**0.009**	**0.008**	**0.04**	**0.03**	**0.03**	**0.04**
	FI (3-0)	3–3 3.0	3–3 3.0	1–3 1.7 ± 0.3	1–3 1.7 ± 0.6	1–3 1.7 ± 0.5	1–3 1.6 ± 0.7	1–3 1.7 ± 0.8	1–3 1.7 ± 0.8	1–3 2.6 ± 0.7	1–3 2.6 ± 0.6	**0.01**	**0.02**	**0.01**	**0.02**	0.05	0.05	0.05	0.05
	MEP Amplitude (µV)	1150–9450 3670.6 ± 122.4	1100–9300 3625.1 ± 118.7	100–5000 1050.0 ±454.2	50–4000 1025.0 ± 136.1	100–6000 1421.3 ± 103.2	100–7000 1410.1 ± 150.3	900–6000 3388.2 ± 114.3	1000–7000 3205.6 ± 105.7	1100–9000 3365.2 ± 403.1	900–11,000 3309.9 ± 485.3	**0.009**	**0.008**	**0.01**	**0.01**	0.05	0.05	0.05	0.05
	MEP Latency (ms)	18.6–22.8 20.8 ± 2.2	18.9–23.0 21.2 ± 1.9	19.8–29.0 24.9 ±3.6	20.1–30.0 24.8 ± 3.3	19.3–31.0 22.6 ± 2.5	17.7–30.0 23.9 ± 4.5	19.0–25.7 22.2 ± 2.1	18.7–25.7 21.1 ± 2.0	18.4–26.3 21.9 ± 2.2	19.3–25.7 22.8 ± 1.7	**0.04**	**0.04**	0.06	0.05	0.06	0.06	0.05	0.05
RA	mcsEMG Amplitude (µV)	450–1950 1097.2 ± 116.3	400–2100 1099.1 ± 170.2	50–500 217.6 ± 63.2	50–500 216.8 ± 69.3	50–600 250.3 ± 67.9	0–600 231.5 ± 69.1	100–1200 405.2 ± 103.4	100–1200 430.2 ± 113.5	100–1800 638.9 ± 69.5	100–2000 692.1 ± 78.2	**0.008**	**0.007**	**0.009**	**0.008**	**0.01**	**0.02**	**0.02**	**0.03**
	FI (3-0)	2–3 2.8 ± 0.4	2–3 2.7 ± 0.3	1–2 1.7 ± 0.4	1–2 1.6 ± 0.5	1–3 1.5 ± 0.6	0–3 1.5 ± 0.5	1–3 2.0 ± 0.4	1–3 2.1 ± 0.6	1–3 2.1 ± 0.7	1–3 2.0 ± 0.6	**0.01**	**0.02**	**0.02**	**0.01**	**0.03**	**0.04**	**0.03**	**0.03**
	MEP Amplitude (µV)	800–2050 1485.4 ± 125.7	750–2000 1434.8 ± 121.3	50–1000 358.2 ± 72.5	50–1000 383.3 ± 43.4	100–600 300.0 ± 45.5	0–600 323.9 ± 94.2	100–1500 500.7 ± 44.8	100–1400 469.9 ± 46.3	100–5000 909.4 ± 140.8	100–5000 959.6 ± 133.2	**0.03**	**0.03**	**0.02**	**0.03**	**0.04**	**0.04**	**0.04**	**0.04**
	MEP Latency (ms)	13.7–15.6 14.8 ± 2.2	13.9–15.9 14.9 ± 2.2	13.2–22.1 17.2 ±3.2	13.6–21.2 17.4 ± 2.7	13.5–20.8 16.7 ± 2.1	0–24.2 16.4 ± 4.7	13.2–19.5 15.6 ± 1.9	13.4–19.0 15.9 ± 1.6	13.2–21.5 17.0 ± 2.7	13.8–22.2 17.4 ± 2.1	**0.04**	**0.04**	0.05	0.05	0.06	0.06	**0.04**	**0.04**
RF	mcsEMG Amplitude (µV)	800–4200 1710.9 ± 105.4	950–4000 1737.2 ± 101.1	50–500 205.0 ± 71.4	50–500 221.6 ± 75.5	0–1500 328.9 ± 83.5	0–1100 273.1 ± 78.4	100–1500 283.6 ± 151.0	0–1500 280.4 ± 150.2	0–4000 407.3 ± 70.1	0–4000 368.3 ± 61.7	**0.009**	**0.008**	**0.01**	**0.01**	**0.01**	**0.02**	**0.01**	**0.02**
	FI (3-0)	2–3 2.9 ± 0.3	2–3 2.8 ± 0.5	1–2 1.5 ± 0.05	1–2 1.5 ± 0.4	0–3 1.2 ± 0.6	0–3 1.3 ± 0.5	1–3 1.5 ± 0.6	0–3 1.4 ± 0.5	0–3 1.4 ± 0.6	0–3 1.4 ± 0.5	**0.009**	**0.007**	**0.008**	**0.007**	**0.009**	**0.008**	**0.009**	**0.008**
	MEP Amplitude (µV)	1050–2250 1527.1 ± 105.6	1000–2100 1499.8 ± 107.5	50–3000 566.6 ± 92.3	50–1800 408.3 ± 87.7	0–1000 300.9 ± 75.2	0–600 323.3 ± 94.5	50–3000 294.8 ± 68.2	0–4000 338.2 ± 91.5	0–2500 414.2 ± 68.7	0–1850 303. ± 76.17	**0.007**	**0.008**	**0.009**	**0.009**	**0.009**	**0.009**	**0.008**	**0.009**
	MEP Latency (ms)	19.1–23.7 21.9 ± 2.0	19.9–23.9 22.0 ± 1.8	18.2–38.4 29.4 ±8.0	17.1–39.3 29.1 ± 7.1	0–43.4 26.2 ± 3.4	0–49.2 28.3 ± 7.1	18.9–32.2 24.7 ± 4.6	0–34.8 23.1 ± 4.7	0–50.5 24.8 ± 9.2	0–56.7 26.6 ± 10.1	**0.03**	**0.03**	**0.04**	**0.04**	0.05	0.05	**0.04**	**0.04**
EXT	mcsEMG Amplitude (µV)	900–3500 1475.6 ± 155.2	850–3000 1423.1 ± 137.1	50–500 268.3 ± 68.4	0–800 230.2 ± 55.8	0–1200 273.7 ± 50.2	0–1100 272.1 ± 31.6	0–3000 347.1 ± 100.1	0–2000 344.3 ± 118.3	0–900 116.6 ± 62.3	0–800 113.4 ± 50.5	**0.009**	**0.008**	**0.009**	**0.007**	**0.01**	**0.02**	**0.008**	**0.009**
	FI (3-0)	3–3 3.0	3–3 3.0	1–2 13 ± 0.5	1–2 1.1 ± 0.4	0–2 1.1 ± 0.5	0–2 1.0 ± 0.4	0–3 1.1 ± 0.6	0–3 1.1 ± 0.6	0–2 0.9 ± 0.3	0–2 0.9 ± 0.6	**0.009**	**0.008**	**0.008**	**0.009**	**0.008**	**0.007**	**0.008**	**0.007**
	MEP Amplitude (µV)	900–1950 1527.4 ± 102.6	850–1800 1411.2 ± 101.1	50–400 141.7 ± 31.3	0–1000 291.6 ± 90.4	0–1000 192.1 ± 72.2	0–700 147.3 ± 32.1	0–3000 263.3 ± 70.2	0–1000 141.2 ± 48.6	0–1000 131.7 ± 37.2	0–1000 126.4 ± 31.8	**0.009**	**0.007**	**0.008**	**0.009**	**0.008**	**0.009**	**0.009**	**0.009**
	MEP Latency (ms)	36.5–40.2 38.4 ± 2.9	37.1–41.1 39.2 ± 2.3	31.2–50.2 43.9 ± 6.0	34–50.3 46.9 ± 8.4	0–52.1 48.5 ± 4.7	0–51.8 47.9 ± 7.2	0–41.2 45.2 ± 6.5	0–49.7 42.5 ± 5.1	0–67.4 47.8 ± 6.5	0–68.1 47.3 ± 8.2	**0.03**	**0.04**	**0.02**	**0.02**	**0.04**	**0.04**	**0.02**	**0.02**
												**Patients** **C3–C5**		**Patients** **C6–Th1**		**Patients** **Th3–Th6**		**Patients** **Th7–L1**	
										**MRI** **injury** **score**	**Side**	**Right**	**Left**	**Right**	**Left**	**Right**	**Left**	**Right**	**Left**
											**Upper** **spinal half**	0–1 0.41 ± 0.2	0.5–1 0.60 ± 0.2	0–1 0.68 ± 0.2	0–1 0.50 ± 0.2	0.5–1 0.66 ± 0.2	0–1 0.47 ± 0.2	0–1 0.84 ± 0.2	0–1 0.68 ± 0.3
											**Lower** **spinal half**	0.5–1 0.58 ± 0.2	0.5–1 0.66 ± 0.25	0–1 0.57 ± 0.2	0–1 0.53 ± 0.3	0.5–1 0.66 ± 0.2	0–1 0.52 ± 0.3	0–1 0.56 ± 0.2	0–1 0.46 ± 0.2

Abbreviations: APB—abductor pollicis brevis muscle; RA–r ectus abdominis muscle; RF—rectus femoris muscle; EXT—extensor digitorum brevis muscle; mcsEMG—sEMG recording during maximal contraction; FI—frequency index (3-0)—frequency of motor units action potentials recruitment during maximal contraction (3—95–70 Hz—normal; 2—65–40 Hz—moderate abnormality; 1—35–10 Hz—severe abnormality; 0—no contraction); MEP—motor-evoked potential; Spinal half injury MRI score (0–1; for each quadrant 1—normal, 0.5—partial injury, 0—total injury); *p* < 0.05 determines significant statistical differences marked in bold.

**Table 4 biomedicines-11-02602-t004:** Spearman’s rank correlations (r_s_) were calculated for the neurophysiological measurements and neuroimaging analysis results in the four groups of patients. Data from the right and left sides or cumulative neuroimaging spinal halves are presented. *p* < 0.05 was assumed as statistically significant for rank correlation. r_s_ values marked bold indicate high or very high correlations.

Parameter	All iSCI Patients					
	**Cumulative mcsEMG amplitude (µV)**					
Cumulative MEP Amplitude (µV)	r_s_			*p*		
	**0.585**			**0.01**		
	Cumulative ENG score					
Cumulative EXT MEP Amplitude (µV)	r_s_			*p*		
	**0.525**			**0.02**		
	**C3–C5 iSCI patients**					
Cumulative left spinal half injury MRI score			Cumulative right spinal half injury MRI score			
Left APB MEP Amplitude (µV)	r_s_	*p*	Right APB MEP Amplitude (µV)		r_s_	*p*
	**0.643**	**0.008**			**0.672**	**0.009**
Left RA MEP Amplitude (µV)	r_s_	*p*	Right RA MEP Amplitude (µV)		r_s_	*p*
	**0.511**	**0.006**			**0.535**	**0.005**
Left RF MEP Amplitude (µV)	r_s_	*p*	Right RF MEP Amplitude (µV)		r_s_	*p*
	**0.625**	**0.005**			**0.692**	**0.001**
Left EXT MEP Amplitude (µV)	r_s_	*p*	Right EXT MEP Amplitude (µV)		r_s_	*p*
	**0.725**	**0.006**			**0.691**	**0.003**
	**C6–Th1 iSCI patients**					
Cumulative left spinal half injury MRI score			Cumulative right spinal half injury MRI score			
Left APB MEP Amplitude (µV)	r_s_	*p*	Right APB MEP Amplitude (µV)		r_s_	*p*
	**0.725**	**0.04**			**0.734**	**0.02**
Left RA MEP Amplitude (µV)	r_s_	*p*	Right RA MEP Amplitude (µV)		r_s_	*p*
	**0.524**	**0.003**			**0.511**	**0.008**
Left RF MEP Amplitude (µV)	r_s_	*p*	Right RF MEP Amplitude (µV)		r_s_	*p*
	**0.922**	**0.02**			**0.823**	**0.03**
Left EXT MEP Amplitude (µV)	r_s_	*p*	Right EXT MEP Amplitude (µV)		r_s_	*p*
	**0.846**	**0.008**			**0.798**	**0.007**
	**Th3–Th6 iSCI patients**					
Cumulative left spinal half injury MRI score			Cumulative right spinal half injury MRI score			
Left APB MEP Amplitude (µV)	r_s_	*p*	Right APB MEP Amplitude (µV)		r_s_	*p*
	0.023	0.16			0.016	0.12
Left RA MEP Amplitude (µV)	r_s_	*p*	Right RA MEP Amplitude (µV)		r_s_	*p*
	**0.826**	**0.006**			**0.725**	**0.008**
Left RF MEP Amplitude (µV)	r_s_	*p*	Right RF MEP Amplitude (µV)		r_s_	*p*
	**0.545**	**0.004**			**0.603**	**0.004**
Left EXT MEP Amplitude (µV)	r_s_	*p*	Right EXT MEP Amplitude (µV)		r_s_	*p*
	**0.622**	**0.009**			**0.611**	**0.008**
	**Th7–L1 iSCI patients**					
Cumulative left spinal half injury MRI score			Cumulative right spinal half injury MRI score			
Left APB MEP Amplitude (µV)	r_s_	*p*	Right APB MEP Amplitude (µV)		r_s_	*p*
	0.051	0.07			0.068	0.07
Left RA MEP Amplitude (µV)	r_s_	*p*	Right RA MEP Amplitude (µV)		r_s_	*p*
	0.043	0.05			0.066	0.06
Left RF MEP Amplitude (µV)	r_s_	*p*	Right RF MEP Amplitude (µV)		r_s_	*p*
	0.325	0.05			0.422	0.05
Left EXT MEP Amplitude (µV)	r_s_	*p*	Right EXT MEP Amplitude (µV)		r_s_	*p*
	**0.528**	**0.008**			**0.622**	**0.007**

Abbreviations: iSCI—incomplete spinal cord injury; MEP—motor-evoked potential recording; APB—abductor pollicis brevis muscle; RA–rectus abdominis muscle; RF–rectus femoris muscle; EXT—extensor digitorum brevis muscle; Spinal half injury MRI score (0–1; for each quadrant 1—normal, 0.5—partial injury, 0—total injury).

## Data Availability

All the data generated or analysed during this study are included in this published article.

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
