# Peer review of "Comparing Parameters of Motor Potentials Recordings Evoked Transcranially with Neuroimaging Results in Patients with Incomplete Spinal Cord Injury: Assessment and Diagnostic Capabilities"

_biomedicines, 2023, doi:10.3390/biomedicines11102602_

Round 1
Reviewer 1 Report
The authors investigated the relationships between different levels and degrees of incomplete spinal cord injury evaluated with magnetic resonance imaging and results of non-invasive electromyography, motor-evoked potentials, and electroneurography. Doing so, they compared MRI and diagnostic neurophysiological test results in 85 iSCI patients lesioned at four distinct levels, performed within 1-2 months after the injury, and 80 matched healthy controls. The study found consistent positive correlations in iSCI patients between the bilateral decrease of the spinal half injury MRI scores and a decrease of transcranially evoked MEPs amplitudes, and revealed significant relationships between the decrease of cumulative mcsEMG and MEP amplitudes and cumulative ENG scores.
All in all, the study is well designed and logically structured. The paper is well written and gives some new interesting results that are of value for iSCI patients.
Nevertheless, there are some points to be improved.
You said: Rare attempts have been made to establish a correlation between the assessment of MRI and MEPs in patients with iSCI, and this correlation has not been previously explored. My question: is this really true?
You say: Open white areas the how the location of bipolar surface electrodes for MEP … - that makes no sense for me.
Question to legend to Fig 1: is the innervation of the motor neuron by the interneurons always excitatory? – My comment: I do not think so.
In the second paragraph of the discussion, more references should be included.
You said: This could imply that more severe structural abnormalities observed in MRI correspond directly to neuronal excitability or responsiveness in the motor system, leading to weaker MEPs. My question: is this astonishing? I think this is well known.
You said: Propriospinal neurons facilitate intersegmental communication within the spinal cord. This coordination can influence motor output patterns. My comment: this is a very interesting aspect. I think it is so interesting for readers that you should prepare a concrete (additional) drawings in which you schematically explain the hypothetical connections of responsible propriospinal neurons (intrinsic spinal cord circuits) for the improved motor output. Ideally, this should be done twice: first for the C3-C5, C6-Th1, and Th3-Th6 subgroups, second for the Th7-L1 group.
You said that “gender, age, height, and weight” of the controls were adjusted to match those of the study group. My comment: in the ms you said nothing about the age spectrum of the patients and also nothing about age-related plasticity potential of the central nervous system.
Reviewer 2 Report
The research article aims to bridge a gap in our understanding of spinal cord injuries and their implications by exploring correlations between MRI scores and neurophysiological metrics. However, upon close examination, there are several critical weaknesses and limitations to consider.
· The study uses a sample size of 85 iSCI patients. While this number might seem adequate at a glance, the subjects are further divided into four categories based on injury location. This breakdown dilutes the sample size in each category, potentially affecting the robustness of the results. Although the inclusion of a control group is commendable, the lack of a comparison with individuals who have complete spinal cord injuries might be seen as a missed opportunity.
· The 0-4 scoring system for evaluating spinal white and grey matter integrity seems arbitrary. The paper doesn’t sufficiently justify or explain the criteria underlying this scoring system, leading to potential questions about its validity.
· Assessments were performed within 1-2 months after the injury. This is a short period, and potential changes in patient condition over time aren’t taken into account. This lack of longitudinal data limits the conclusions that can be drawn.
· Throughout the paper, there is a heavy emphasis on the correlation coefficients. While correlations are useful, they merely describe the relationship between two variables and cannot ascertain causation. The paper seems to blur this distinction, especially when discussing the implications of the findings.
· The study doesn’t sufficiently account for potential confounding variables. Factors such as pre-injury fitness, other associated injuries, and patient demographics might all influence outcomes and should have been considered or controlled for.
· The findings in the Th7-L1 group that present weak and statistically insignificant correlations deserve more scrutiny and discussion. Why might this subgroup differ from the others? This is a significant limitation that isn’t thoroughly explored.
· The conclusions draw strong recommendations based on correlations. As aforementioned, correlations don’t infer causation. The potential implications for patient care based on these findings might be premature.
· The study emphasizes that each patient’s response to injury is unique. However, the paper tries to generalize findings to broader groups. There’s a tension between these two assertions that isn’t resolved.
Recommendations for Improvement:
· Future studies should consider larger, more balanced sample sizes, introduce additional comparison groups, and develop a more standardized MRI scoring system after rigorous validation.
· The authors should be clear about the difference between correlation and causation and temper conclusions accordingly.
· Investigate other potential variables, including detailed patient demographics, pre-injury health status, and the nature of the injuries themselves.
· A longitudinal study capturing changes over time would provide a more holistic understanding of recovery patterns.
Reviewer 3 Report
A very interesting study involving 85 patients with incomplete spinal cord injury and 80 healthy controls.
The study tries to shed light on the complex interactions between spinal cord injury levels/severity, the structural damage, the neurophysiological measures, and outcomes of motor function. The authors used numerous methods for evaluation, a) MEP (motor-evoked potentials), b) EMG (electromyography), and c) ENG (electroneurography) to reveal possible correlations between MEP amplitudes and MRI injury scores.
The study involved subjects 1-2 months after injury and as mentioned healthy controls. There are no differences between these two groups in terms of gender, weight, height and age, thus the two groups may be considered as balanced in terms of their population and their characteristics.
In general the study is well designed and presented and found no linguistic issues.
In my opinion can be accepted with minor changes.
1. I would like to see a few numeric results in the abstract that is already large enough.
2. It is mentioned that the control group was used to establish reference values. However in the manuscript there are not produced reference values (reference values in laboratory settings are usually the values between 2.5-98.5 percentiles of large >150 population). Perhaps the term reference values is not appropriate, however cannot propose another term!.
Author Response
Dear Reviewer 3,
We hope this message finds you well, and we want to express our sincere gratitude for taking the time to review our manuscript. Your insightful comments have been immensely valuable in improving the quality of the paper. We would also like to mention that several of the revisions (marked with different colours) made were in response to suggestions from other reviewers.
In our response, we will address each of your comments individually, and to make it easier for you to identify the changes, we have highlighted them in blue in the revised manuscript. Your feedback is highly appreciated, and we are committed to ensuring that all your concerns and suggestions are thoroughly addressed.
Comment 1:
I would like to see a few numeric results in the abstract that is already large enough.
Response 1:
Thank you. We have added some numeric results in the abstract.
Comment 2:
It is mentioned that the control group was used to establish reference values. However in the manuscript there are not produced reference values (reference values in laboratory settings are usually the values between 2.5-98.5 percentiles of large >150 population). Perhaps the term reference values is not appropriate, however cannot propose another term!
Response 2:
Thank you. We’ve had the same problem while preparing the written version of the manuscript, therefore in the text we have tried to use more often the term “control group” than reference values. We hope that this will not disturb the reader with the main points of the work. These findings in the healthy are only so that the reader can trace the difference in the neurophysiological status of the healthy from those with different levels of spinal cord damage.
Thank you once again for your valuable input, and we look forward to sharing the revised manuscript with you shortly.
Best regards,
Authors
Reviewer 4 Report
Dear Author
This study is useful for establishing an evaluation method regarding iSCI. On the other hand, the length of each section may not convey the importance of this study to the reader. Therefore, the following points should be revised.
● The abstract is too long and should describe this study in a simple manner in accordance with the submission rules. The introduction also lists the epidemiology of iSCI and issues related to treatment, but there are few sections describing the importance of this evaluation. The introduction should clarify the significance of this study.
● Amplitude in table3 should be shown as median to make the table easier to read.
Minor points
・Add BMI and disease duration to Table 1. The description should be minimum - maximum.
Author Response
Dear Reviewer,
We hope this message finds you well, and we want to express our sincere gratitude for taking the time to review our manuscript. Your insightful comments have been immensely valuable in improving the quality of the paper. We would also like to mention that several of the revisions (marked with different colours) made were in response to suggestions from other reviewers.
In our response, we will address each of your comments individually, and to make it easier for you to identify the changes, we have highlighted them in yellow in the revised manuscript. Your feedback is highly appreciated, and we are committed to ensuring that all your concerns and suggestions are thoroughly addressed.
Comment 1:
The abstract is too long and should describe this study in a simple manner in accordance with the submission rules. The introduction also lists the epidemiology of iSCI and issues related to treatment, but there are few sections describing the importance of this evaluation.
Response 1:
Thank you for your observation. You are correct. We have indeed condensed the abstract significantly and relocated certain pertinent points from it to the introduction section. We hope this adjustment allows us to emphasize and elucidate the significance of this evaluation more effectively.
Comment 2:
Amplitude in table 3 should be shown as median to make the table easier to read.
Response 2:
It's not feasible since it's a numerical value, rendering it impossible to conduct statistical comparisons, as based on our statistical knowledge.
Minor points
Comment 3:
Add BMI and disease duration to Table 1. The description should be minimum - maximum.
Response 3:
Ok, we have added it. The disease duration is expressed as the average time from injury (in months).
Thank you once again for your valuable input, and we look forward to sharing the revised manuscript with you shortly.
Best regards,
Authors
Round 2
Reviewer 1 Report
Dear authors,
thank you for the clarifications and the improvement of the manuscript.
Reviewer 2 Report
The authors have improved the manuscript while addressing all my comments in a satisfactory manner. I have no further comments and recommend to accept.
Reviewer 4 Report
I have reviewed the revised manuscript again. There are no comments from me on this manuscript.
Thank you for your revisions.